# A Self-enhancement Multitask Framework for Unsupervised Aspect Category Detection

**Thi-Nhung Nguyen[1], Hoang Ngo[1], Kiem-Hieu Nguyen[2], Tuan-Dung Cao[2]**
[1]VinAI Research
[2]School of Information and Communications Technology,
Hanoi University of Science and Technology
{v.nhungnt89,v.hoangnv49}@vinai.io, {hieunk,dungct}@soict.hust.edu.vn

## Abstract

Our work addresses the problem of unsupervised Aspect Category Detection using a small set of seed words. Recent works have focused on learning embedding spaces for seed words and sentences to establish similarities between sentences and aspects. However, aspect representations are limited by the quality of initial seed words, and model performances are compromised by noise. To mitigate this limitation, we propose a simple framework that automatically enhances the quality of initial seed words and selects high-quality sentences for training instead of using the entire dataset. Our main concepts are to add a number of seed words to the initial set and to treat the task of noise resolution as a task of augmenting data for a low-resource task. In addition, we jointly train Aspect Category Detection with Aspect Term Extraction and Aspect Term Polarity to further enhance performance. This approach facilitates shared representation learning, allowing Aspect Category Detection to benefit from the additional guidance offered by other tasks. Extensive experiments demonstrate that our framework surpasses strong baselines on standard datasets.

## 1 Introduction

Aspect Category Detection (ACD), Aspect Term Extraction (ATE), and Aspect Term Polarity (ATP) are three challenging sub-tasks of Aspect Based Sentiment Analysis, which aim to identify the aspect categories discussed (e.i., FOOD), all aspect terms presented (e.i., 'fish', 'rolls'), and determine the polarity of each aspect term (e.i., 'positive', 'negative') in each sentence, respectively. To better understand these problems consider the example in Table 1.

Unsupervised ACD has mainly been tackled by clustering sentences and manually mapping these clusters to corresponding golden aspects using top representative words (He et al., 2017; Luo et al.,

2019; Tulkens and van Cranenburgh, 2020; Shi et al., 2021). However, this approach faces a major problem with the mapping process, requiring manual labeling and a strategy to resolve aspect label inconsistency within the same cluster. Recent works have introduced using seed words to tackle this problem (Karamanolakis et al., 2019; Nguyen et al., 2021; Huang et al., 2020). These works direct their attention to learning the embedding space for sentences and seed words to establish similarities between sentences and aspects. As such, aspect representations are limited by a fixed small number of the initial seed words. (Li et al., 2022) is one of the few works that aims to expand the seed word set from the vocabulary of a pre-trained model. However, there is overlap among the additional seed words across different aspects, resulting in reduced discriminability between the aspects. Another challenge for the unsupervised ACD task is the presence of noise, which comes from out-of-domain sentences and incorrect pseudo labels. This occurs because data is often collected from various sources, and the pseudo labels are usually generated using a greedy strategy. Current methods handle noisy sentences by either treating them as having a GENERAL aspect (He et al., 2017; Shi et al., 2021) or forcing them to have one of the desired aspects (Tulkens and van Cranenburgh, 2020; Huang et al., 2020; Nguyen et al., 2021). To address incorrect pseudo labels, (Huang et al., 2020; Nguyen et al., 2021) attempt to assign lower weights to uncertain pseudo-labels. However, these approaches might still hinder the performance of the model as models are learned from a large amount of noisy data.

In this paper, we propose A Self-enhancement Multitask (ASeM) framework to address these limitations. Firstly, to enhance the understanding of aspects and reduce reliance on the quality of initial seed words, we propose a *Seedword Enhancement Component* (SEC) to construct a high-quality set

| {While the fish$_{[pos]}$ is unquestionably fresh, rolls$_{[neg]}$ tend to be inexplicably bland.}$_{\textbf{FOOD}}$ |
| --- |

Table 1: Aspect Category Detection, Aspect Term Extraction, and Aspect Term Polarity example.

of seed words from the initial set. The main idea is to add keywords that have limited connections with the initial seed words. Connections are determined by the similarity between the keyword's context (the sentence containing the keywords) and the initial seed words. In this way, our task is simply to find sentences with low similarity to the initial seed words and then extract their important keywords to add to the seed word set. Since pseudo-label generation relies on the similarity between sentences and seed words, we expect that the enhanced seed word set would provide sufficient knowledge for our framework to generate highly confident pseudo-labels for as many sentences as possible. Secondly, to reduce noise in the training data, instead of treating them as having a GENERAL aspect (He et al., 2017; Shi et al., 2021) or forcing them to have one of the desired aspects (Tulkens and van Cranenburgh, 2020; Huang et al., 2020; Nguyen et al., 2021), we propose to leverage a retrieval-based data augmentation technique to automatically search for high-quality data from a large training dataset. To achieve this, we leverage a *paraphrastic encoder* (e.g. Arora et al. (2017); Ethayarajh (2018); Du et al. (2021)), trained to output similar representations for sentences with similar meanings, to generate sentence representations that are independent of the target task (ACD). Then, we generate task embeddings by passing the prior knowledge (e.g., seed words) about the target task to the encoder. These embeddings are used as a query to retrieve similar sentences from the large training dataset (data bank). In this way, our framework aims to effectively extract domain-specific sentences that can be well understood based on seed words, regardless of the quality of the training dataset.

Another contribution to our work concerns the recommendation of multi-tasking learning for unsupervised ACD, ATE and ATP in a neural network. Intuitively, employing multi-task learning enables ACD to leverage the benefits of ATE and ATP. Referring to the example in Figure 1, ATE extracts Opinion Target Expressions (OTEs): *'fish'* and *'rolls'*, which requires understanding the emotion *'positive'* (expressed as *'unquestionably fresh'*) for *'fish'* and *'negative'* (expressed as *'inexplica-*

*bly bland'*) for *'rolls'*. Meanwhile, ACD wants to detect the aspect category of the sentence, requiring awareness of the presence of "fish" and "rolls" (terms related to the aspect of "food") within the sentence. Despite the usefulness of these relationships for ACD, the majority of existing works do not utilize this information. (Huang et al., 2020) is one of the few attempts to combine learning the representations of aspect and polarity at the sentence level before passing them through separate classifiers.

Our contributions are summarized as follows:

- We propose a simple approach to enhance aspect comprehension and reduce the reliance on the quality of initial seed words. Our framework achieves this by expanding the seed word set with keywords, based on their uncertain connection with the initial seed words.

- A retrieval-based data augmentation technique is proposed to tackle training data noise. In this way, only data that connect well with the prior knowledge (e.g., seed words) about the target task is used during training. As a result, the model automatically filters out-of-domain data and low-quality pseudo labels.

- We propose to leverage a multi-task learning approach for unsupervised ACD, ATE, and ATP, aiming to improve ACD through the utilization of additional guidance from other tasks during the shared representation learning.

- Comprehensive experiments are conducted to demonstrate that our framework outperforms previous methods on standard datasets.

## 2   Related Works

Topic models were once the dominant approach (Brody and Elhadad, 2010; Mukherjee and Liu, 2012; Chen et al., 2014) for unsupervised Aspect Category Detection. However, they can produce incoherent aspects. Recently, neural network-based methods have been developed to address this challenge.

**Cluster Mapping-based resolvers:** These methods utilize neural networks to cluster effectively

and manually map (many-to-many mapping) the clusters to their corresponding aspects. They employ attention-based autoencoders (He et al., 2017; Luo et al., 2019) or contrast learning approach (Shi et al., 2021) for clustering. Shi et al. (2021) further enhance performance by using knowledge distillation to learn labels generated after clustering.

**Seed words-based resolvers:** These approaches automate the aspect category mapping process by utilizing seed words that indicate aspect appearance. Angelidis and Lapata (2018) use the weighted sum of seed word representations as aspect representations, allowing mapping one-to-one in the auto-encoder model. Recent works focus on learning embedding spaces for sentences and seed words, generating pseudo labels for weakly supervised learning. They use Skip-gram (Mikolov et al., 2013) for embedding space learning and convolutional neural networks or linear layers for classification (Huang et al., 2020; Nguyen et al., 2021). Huang et al. (2020) jointly learn ACD with Sentence-level ATP, while Nguyen et al. (2021) consider the uncertainty of the initial embedding space. Without any human supervision, (Tulkens and van Cranenburgh, 2020; Li et al., 2022) rely solely on label names, similar to seed words. Tulkens and van Cranenburgh (2020) detect aspects using cosine similarity between pre-trained aspect and label name representations, while Li et al. (2022) train the clustering model with instance-level and concept-level constraints.

## 3 Method

Our framework addresses three tasks for which no annotated data is available: Aspect Category Detection (ACD), Aspect Term Extraction (ATE), and Aspect Term Polarity (ATP). ACD involves assigning a given text to one of K pre-defined aspects of interest. ATE extracts OTEs in the text. ATP assigns a sentiment to each OTE. Note that, during training, we do not use any human-annotated samples, but rather rely on a small set of seed words to provide supervision signals.

Our framework called **ASeM** (short for **A S**elf-**e**nhancement **M**utitask Framework), consists of three key components: (i) Pseudo-label generation, (ii) Retrieval-based data augmentation, and (iii) Classification. Figure 1 presents an overview of the framework. Initially, we extract a small subset of the training data to serve as the task-specific in-domain data. Based on the quality of the initial

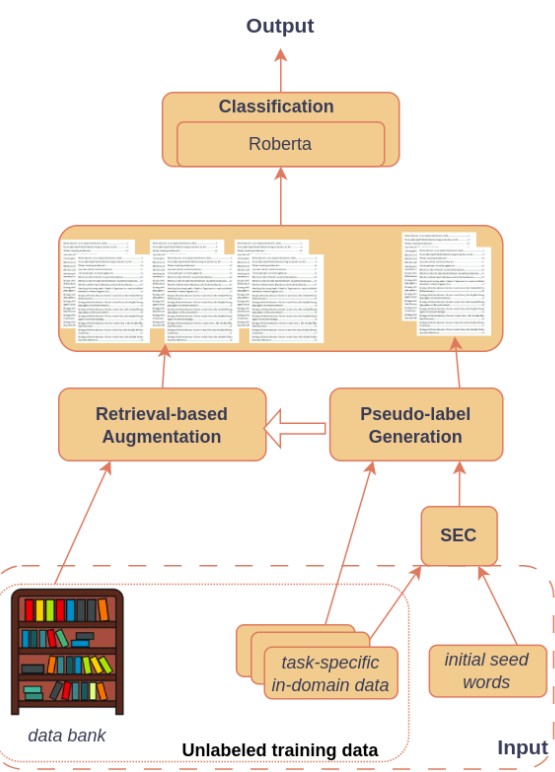

Figure 1: Overview of our proposed Self-enhancement Multitask (ASeM) framework.

seed words in this dataset, we utilize SEC to expand the set of seed words in order to enhance its quality. By feeding the task-specific in-domain data and enhanced seed words to the pseudo-label generation, we obtain high-quality pseudo labels for the task-specific in-domain data. Then, we leverage the retrieval-based augmentation to enhance the number of training samples from the data bank (the remaining part of the training data), based on our prior knowledge of the target task (seed words, task-specific in-domain data with high-quality pseudo labels). To this end, the high-quality pseudo labels and augmented data are passed through a multitask classifier to predict the task outputs.

### 3.1 Pseudo-label generation

The first step in our framework is to generate pseudo-labels for the three subtasks ACD, ATE, and ATP, on a small unannotated in-domain dataset. In detail, the pseudo-labels for the tasks are created as follows:

**Aspect Category Detection:** First, we map dictionary words into an embedding space by training CBOW (Mikolov et al., 2013) on the unlabeled training data. Second, we embed sentences from the task-specific in-domain data as

$\boldsymbol{s} = sum(\boldsymbol{w}_1, \boldsymbol{w}_2, .., \boldsymbol{w}_n)$, in which $\boldsymbol{w}_i$ is the representation of the $i^{\text{th}}$ word and $n$ is the sentence length. Similarly, the aspect category representation $\boldsymbol{a}_i = sum(\boldsymbol{w}_{i,1}^{(a)}, \boldsymbol{w}_{i,2}^{(a)}, .., \boldsymbol{w}_{i,l_i}^{(a)})$, in which $\boldsymbol{w}_{ij}^{(a)}$ is the representation of the $j^{\text{th}}$ seed word of the $i^{\text{th}}$ aspect, and $l_i$ is the number of seed words in the $i^{\text{th}}$ aspect. To this end, aspect category pseudo label of a sentence $s$ is defined as follows:

$$y = \underset{i}{argmax}(sim(s, a_i)), 1 \le i \le K \quad (1)$$

where $sim(s, a_i)$ is the similarity between sentence $s$ and aspect $a_i$. Given set $G_{a_i} = T_{a_i} \cup H_{a_i}, 1 \le i \le K$ in which $H_{a_i}$ is the set of given initial seed words, $T_{a_i}$ is the set of additional seed words. The similarity is calculated as follows:
**if** $s' \cap G_{a_i} = \varnothing, \forall 1 \le i \le K$ **then**
$sim(s, a_i) = \mathbf{s}^T \mathbf{a}_i, \forall 1 \le i \le K$
**else**

$$sim(s, a_i) = \begin{cases} \sum\limits_{w \in s' \cap G_{a_i}} \mathbf{w}^T \mathbf{s}, & \text{if } s' \cap G_{a_i} \neq \varnothing \\ 0, & \text{otherwise} \end{cases}$$

where $\mathbf{s}$ and $\mathbf{a}_i$ are sentences and aspect representations, respectively. $w$ and $\mathbf{w}$ are a word in a sentence and its representation. $s'$ is the set of words in the sentence $s$.

As discussed in the introduction, our framework proposes SEC as described in Algorithm 1 to obtain $T_{a_i}$. To begin, we generate temporary pseudo labels for all given sentences using the initial seed words. Based on the obtained pseudo-labels, we extract nouns and adjectives (called *keywords*) in the sentences for each aspect label and then extract keywords that appear in multiple aspects (called *boundary keywords*) and obtained $T_b$. At line 6, we calculate the connection between sentences and initial seed words based on the difference between the similarity of the sentence with its two most similar aspects. Note that, if $Connection(s) \ge \gamma$, in which $\gamma$ is a hyper-parameter, sentences $s$ are considered to have a certain connection with seed words, and if $Connection(s) < \gamma$, there are uncertain connections. At line 12, we extract keywords from the sentences with uncertain connections and obtain $T_u$. Finally, the intersection of $T_b$ and $T_u$ will be mapped to the relevant aspect. We utilize a variant of *clarity scoring function* (Cronen-Townsend et al., 2002) for the automatic mapping. *Clarity* measures the likelihood of observing a word $w$ in the subset of sentences related to aspect $a_i$, as

---

**Algorithm 1:** Seedword Enhancement Component (SEC)

**Input:** sentence set $S$, initial seed word set $H$, threshold $\gamma$
**Output:** additional seed word set $T_a$

1 **begin**
2      $P \leftarrow$ Pseudo-Label-Generation$(S, H)$ with Eq. 1;
3      $T_b \leftarrow$ Boundary-Keywords-Extraction$(S, P)$ ;
4      $S_u \leftarrow \varnothing$     ▷ Initialize set of sentences with uncertain pseudo-label;
5      **for** $s \in S$ **do**
6          Calculate $Connection(s)$ ;
7          **if** $Connection(s) < \gamma$ **then**
8             Add $(s)$ to $S_u$;
9          **end**
10      **end**
11      $P_u \leftarrow$ Pseudo label of $S_u$;
12      $T_u \leftarrow$ Keywords-Extraction$(S_u, P_u)$   ▷ Extract keywords from sentences with uncertain prediction;
13      $T \leftarrow T_b \cap T_u$ ;
14      $T_a \leftarrow$ Auto mapping$(T)$;
15      **return** $T_a$;
16 **end**

---

compared to $a_j$. A higher score indicates a greater likelihood of word w being related to aspect $a_i$.

$$clarity_{(a_i, a_j)}(w) = t_{a_i}(w) log \frac{t_{a_i}(w)}{t_{a_j}(w)} \quad (2)$$

where $t_{a_i}(w)$ and $t_{a_j}(w)$ correspond to the $l_1$-normalized TF-IDF scores of $w$ in the sentences annotated pseudo-label with aspect $a_i$ and $a_j$, respectively.

In the training process, after obtaining pseudo labels, SEC recalculates the certainty of connections similar to lines 5 to 10 of Algorithm 1, then removes uncertain connections $S_u$ out of $S$.

**Aspect Term Extraction:** We extract aspect terms by considering all nouns that appear more than $m$ times in the corpus.

**Aspect Term Polarity:** After generating aspect term pseudo-labels, we find polarity pseudo-labels of terms based on the context window around them. In detail, the generation will be carried out similarly to the ACD subtask with the input being the context window and polarity seed words.

## 3.2 Retrieval-based Data Augmentation

To select high-quality data from an unannotated data bank containing noise, we select sentences with content similar to the reliable knowledge we have about the task-specific in-domain data, for example, seed words/a small in-domain dataset having certain connections with seed words. To do this, we first utilize a *paraphrastic sentence encoder* to create representations for sentences in the data bank and the target task. The task embedding will be used as a query to find high-quality sentences in the sentence bank. Figure 2 illustrates our retrieval-based data augmentation. The sentence encoder, task embedding, and data retrieval process occur as follows:

**Sentence Encoder:** We leverage a *paraphrastic encoder* to generate similar representations for semantically similar sentences. In detail, the encoder is a Transformer pre-trained with masked language modeling (Kenton and Toutanova, 2019; Conneau and Lample, 2019), it is finetuned by a triplet loss $L(x, y) = max(0, \alpha - cos(x, y) + cos(x, y_n))$ on paraphrases from Natural Language Inference entailment pairs (Williams et al., 2018), Quora Question Pairs, round-trip translation (Wieting and Gimpel, 2018), web paraphrases (Creutz, 2018), OpenSubtitles (Lison et al., 2018), and Europarl (Koehn, 2005) to maximize cosine similarity between similar sentences. Positive pairs $(x, y)$ are either paraphrases or parallel sentences (Wieting et al., 2019), and the negative $y_n$ is selected to be the hardest in the batch.

**Task embedding:** The task embedding uses a shared paraphrastic encoder with sentence embeddings, which is used to embed the prior knowledge about the target task (seed words, task-specific in-domain data with high-quality pseudo labels). In this task, given the prior knowledge, each representation of a seed word or sentence in the prior knowledge is considered a representation of the target task.

**Unsupervised Data Retrieval:** We use task embedding as queries to retrieve a subset of the large sentence bank. For each task embedding, we select $k$ nearest neighbors based on cosine similarity.

## 3.3 Classification

In this component, we train a neural network of multitask learning ATE, ATP, and ATE. We expect that multitask learning can provide additional guidance for ACD from other tasks during the

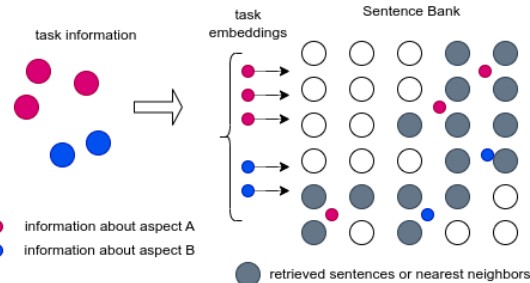

Figure 2: Retrieval-based augmentation ($k = 3$). Information for each aspect is derived from prior knowledge, including seed words and task-specific in-domain sentences having certain connections with seed words.

shared representation learning, resulting in improved ACD performance. In detail, given an input sentence $W = [w_1, ...w_n]$ with n words, we employ a pretrained language model, e.g. RoBERTa (Liu et al., 2019) to generate a shared sequence of contextualized embedding $\mathbf{W} = [\mathbf{w}_1, ..., \mathbf{w}_n]$ for the words (using the average of hidden vectors for word-pieces in the last layer of the pretrained language model). Then, for each task $t_i \in \{ACD, ATE, ATP\}$, we feed the vector sequence $\mathbf{W}$ into a feed-forward network to find the corresponding probability scores $p_W^{t_i}, \forall t_i \in \{ACD, ATE, ATP\}$. Note that, we treat the ACD problem as a sentence classification problem, while ATE and ATP are treated as BIO sequence labeling problems.

Given the training dataset $X_l$, which consists of task-specific in-domain data and augmented data, along with their corresponding pseudo labels $y_l^t$ generated by Pseudo label Generation. The loss function is defined as follows:

$$L^T = -\frac{1}{|X_l|} \sum_{t_i} \sum_{x_l \in X_l} \sum_{c \in C_{t_i}} \lambda_{t_i} y_l^{t_i} log(p_{l,c}^{t_i}) \quad (3)$$

in which $C_{t_i}$ is the set of labels of task $t_i$ and $\lambda_i$ is a hyperparameter determining the weight of task $t_i$.

## 4 Experiments

### 4.1 Datasets

Following previous works (Huang et al., 2020; Shi et al., 2021; Nguyen et al., 2021) we conduct experiments on three benchmark datasets with different domains as described in Table 2.

**Restaurant/Laptop** containing reviews about restaurant/laptop (Huang et al., 2020). We use the

training dataset as the sentence bank and the SemEval training set (Pontiki et al., 2015, 2016) as task-specific in-domain data and dev set with a ratio of 0.85. The test set is taken from SemEval test set (Pontiki et al., 2015, 2016). Restaurant contains labeled data for all three tasks, where ACD has five aspect category types and ATP has two polarity types. Laptop only has labels for ACD with eight aspect category types, and Sentence-level ATP with two polarity types. For initial seed words, following (Huang et al., 2020; Nguyen et al., 2021) we have five manual seed words and five automatic seed words for each label of ACD and Sentence-level ATP.

**CitySearch** containing reviews about restaurants (Ganu et al., 2009), in which the test set only contains labeled ACD data with three aspect category types. Similarly to Restaurant/Laptop, we use SemEval training set (Pontiki et al., 2014, 2015, 2016) as task-specific in-domain data and dev set with a ratio of 0.85. For seed words, similar to (Tulkens and van Cranenburgh, 2020), for Citysearch we use the aspect label words *food, ambience, staff* as seed words.

Following previous works, we remove the multi-aspect sentences from all datasets and only evaluate ATE and ATP on sentences with at least one OTE; multi-polarity sentences are removed when evaluating Sentence-level ATP.

| Dataset | Train | SemEval | Test |
|---|---|---|---|
| Citysearch | 279,862 | 1474 | 1490 |
| Restaurant | 17,027 | 855 | 694 |
| Laptop | 14,683 | 476 | 307 |

Table 2: Statistics of the datasets.

### 4.2 Hyper-Parameters

For the learning framework, the best parameters and hyperparameters are selected based on the validation sets. For pseudo label generation, we use nltk[1] for tokenizing and Gensim[2] to train CBOW with embedding size of 200 and number of epochs of 10, the window of 10, negative sample size of 5. The connection measure $\gamma$ is tuned within the range of $[0, 700]$, ATP context window size is tuned within the range of $[20, 100]$, and $m$ is set to 2. For the ATE task, we use *nltk* for POS tagging and

---

[1] https://www.nltk.org/
[2] https://radimrehurek.com/gensim/

*textblob*[3] to extract nouns, noun phrases and adjectives. For retrieval-based data augmentation, we select: *paraphrastic encoder* pretrained by Du et al. (2021), and tune the number of nearest neighbors $k$ in the range of [1, 20] with step 1. For the classification network, we use Roberta-large (Liu et al., 2019) with batch size 16, learning rate 1e-5, AdamW optimizer, weight decay 1e-5, task weights $\lambda_{t_i}$ is set to 1, 0.8, 0.6 for ACD, ATE, and ATP respectively. We report the average performances of five different runs with random seeds.

### 4.3 Baselines

We compare ours to recent unsupervised ACD methods, including **ABAE** (He et al., 2017), **CAt** (Tulkens and van Cranenburgh, 2020), **JASen** (Huang et al., 2020), **SSCL** (Shi et al., 2021), **UCE** (Nguyen et al., 2021), and **PCCT** (Li et al., 2022). Currently, **PCCT** is state-of-the-art on all three datasets.

Concerning the utilization of seed words to guide inference for larger LLMs, such as generative models like GPT-3.5, we conducted an experiment involving model `gpt-3.5-turbo` (from OpenAI) to infer aspect labels. The prompt used is described in detail in section A.

## 5 Evaluation

First, we report the results of our framework on the ACD task in Table 3, and ATP and ATE in Table 4. During training, we utilize multitask learning for ACD, ATP, and ATE. However, due to limited labeled tasks in the test datasets, we only evaluate tasks with labels. Further details can be found in subsection 4.1. The results of prior methods were collected from the respective works. For initial seed words, we employ the seed words recommended by (Huang et al., 2020) for the Restaurant and Laptop, as well as the seed words suggested by (Tulkens and van Cranenburgh, 2020) for City-Search. Following (Huang et al., 2020), we evaluate the ACD performance by accuracy and macro-F1. Similarly, we evaluate ATP(s) (shorted for Sentence-level ATP) and ATP (shorted for Term-level ATP) by accuracy and macro-F1; and ATE by F1-score. As can be seen, despite using less human supervision compared to manual mapping-based methods, seed word-based methods yield competitive results.

---

[3] https://textblob.readthedocs.io/

| Method | Restaurant | | Laptop | | Citysearch | |
|---|---|---|---|---|---|---|
| | Acc | macro-F1 | Acc | macro-F1 | Acc | macro-F1 |
| Methods with Manual Mapping | | | | | | |
| ABAE (2017) | 67.3 | 45.3 | 59.8 | 56.2 | 85.7 | 77.5 |
| SSCL (2021) | - | - | - | - | 89.7 | 87.0 |
| Methods with Aspect Name/Seed Words | | | | | | |
| CAt (2020) | 66.3 | 46.2 | 58.0 | 58.6 | 83.6 | 82.5 |
| JAsen (2020) | 83.8 | 66.3 | 71.0 | 69.7 | 87.3 | 86.2 |
| UCE (2021) | 83.1 | 66.1 | 71.3 | 71.3 | - | - |
| PCCT (2022) | 85.3 | 79.2 | 74.3 | 73.4 | 90.6 | 89.8 |
| ASeM | **90.0** | **82.0** | **76.5** | **75.7** | **92.2** | **90.0** |

Table 3: The performance of ASeM on Aspect Category Detection.

| Method | Restaurant | | | Laptop |
|---|---|---|---|---|
| | ATP(s) | ATP | ATE | ATP(s) |
| JASen | 79.4 | - | - | 74.6 |
| ASeM | **80.5** | **33.7** | **48.2** | **78.4** |

Table 4: The performance of ASeM on Aspect Term Polarity (ATP), Sentence-level ATP (ATP(s)) and Aspect Term Extraction (ATE). In our work, Sentence-level ATP is not directly learned, but rather its labels are inferred based on the polarity labels of terms in that sentence.

The results compared with GPT-3.5 are reported in Table 5. In detail, while GPT-3.5 performs well in terms of food, drinks, and service aspects and demonstrates superior accuracy, our model outperforms in terms of location and ambience aspects as well as overall macro-F1 score. It is not surprising that GPT-3.5 demonstrates strong performance on this dataset due to its immense underlying knowledge base. However, when considering factors like scalability, computational demands, complexity, and inference time, our method exhibits competitive results.

Overall, ASeM demonstrates state-of-the-art performance in Aspect Category Detection across various domains, providing clear evidence of the effectiveness of the proposed framework.

**Ablation Study**: To investigate the impact of each proposed component for ASeM, we evaluate our ablated framework over the Restaurant dataset. Table 6 allows us to assess the contribution of each proposed component to the overall performance of ASeM.

*W/o Seedword Enhanment Component*. Ablating the SEC includes eliminating the search for additional seed words $T_a$ and then assigning $T_a = \emptyset$. It

can be observed that removing SEC significantly impairs the accuracy of ACD compared to other tasks, clearly demonstrating the benefits of expanding seed words for ACD.

*W/o Retrieval-based Data Augmentation*. We ignore data augmentation and follow previous works Huang et al. (2020); Nguyen et al. (2021) by training our classification on the entire training data. As presented in the table, the *-aug* model exhibits a significant performance decrease compared to ASeM, particularly in 3 out of 4 tasks where it performs the worst among the ablated versions. This clearly demonstrates the adverse impact of noise on the framework's accuracy and the limitations it imposes on the contributions of other components within the framework, as well as the effectiveness of the proposed method in addressing noise.

*W/o Multi-task Learning*. Next, we ablate multi-task learning and train the three tasks (ACD, ATP,

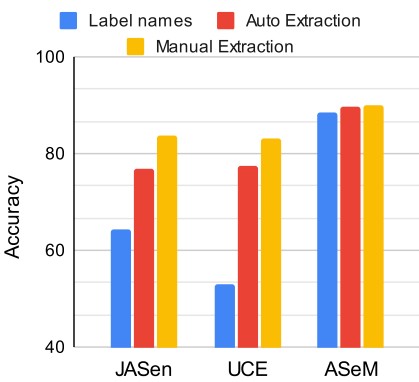

Figure 3: ACD performance of seed words-based methods with different initial seed word sets. *Label names* is derived from the aspect name (e.g. 'food'), *Auto extraction* is automatically extracted from a small labeled dataset (Angelidis and Lapata, 2018), *Manual extraction* is manually extracted by experts (Huang et al., 2020).

| Method | FOOD F1 | DRINKS F1 | SERVICE F1 | LOCATION F1 | AMBIENCE F1 | Overall Acc | Overall macro-F1 |
|---|---|---|---|---|---|---|---|
| ASeM | 93.4 | 76.2 | 87.7 | **63.2** | **88.4** | 90.0 | **82.0** |
| GPT-3.5 | **94.9** | **84.7** | **91.2** | 45.5 | 85.0 | **91.4** | 80. |

Table 5: Performance of ASeM vs. GPT-3.5 on Restaurant Dataset.

| Methods | ACD | ATP(s) | ATP | ATE |
|---|---|---|---|---|
| ASeM | 90.0 | 80.5 | 33.7 | 48.2 |
| -SEC | 86.3 | 79.7 | 31.7 | 46.6 |
| -aug | 86.3 | 74.4 | 28.9 | 46.7 |
| -joint | 88.0 | 79.3 | 32.4 | 42.8 |

Table 6: Ablation study

ATE) separately. While inflicting less damage to ACD compared to other ablated versions, the *-joint* exhibits substantial performance benefits for all component tasks through the straightforward approach of joint learning. It results in a 2% increase in accuracy for ACD, a 1.3% boost in F1-score for ATP, and a remarkable 5.4% improvement in F1-score for ATE.

In general, the elimination of any component from ASeM would significantly hurt the performance, clearly demonstrating the benefits of the proposed components.

# 6 Analysis

In this section, we thoroughly analyze the performance of our framework concerning the quality of seed words and training data.

## 6.1 The quality of seed words

Firstly, we examine the effect of the **initial seed words** on the performance of our framework. Figure 3 shows the accuracy of our framework using different initial seed word sets on the Restaurant dataset, comparing them with other state-of-the-art methods based on seed words. As can be seen, both JASen and UCE exhibit significant variations in performance when changing the initial seed word sets, highlighting their strong dependence on the quality of initial seed words. Meanwhile, ASeM demonstrates a good adaptation ability to the quality of initial seed words, delivering promising results across all three sets of initial seed words. Secondly, we discuss the effectiveness of the **additional seed words** generated by SEC, compared to other methods of adding seed words. The results are reported in Figure 4. While baselines fluctuate, SEC consistently

improves. We suspect that the decline in ACD performance may be attributed to the mismapping of seed words (e.g. 'music' mapped to the aspect 'drink' instead of 'service') and contextual ambiguity when different aspect seed words co-occur (e.g. 'we ended our great experience by having gulab jamun dessert$_{[FOOD]}$ recommended by the waiter$_{[SERVICE]}$'). SEC selectively adds words less connected with seed words, thereby reducing conflicts and minimizing the effect of mismapping. In addition, SEC aids in identifying previously unextracted aspect terms (e.g. *martinis, hot dogs*), as evidenced by the improved ATE performance, providing insight into how SEC enhances the performance of ACD.

## 6.2 Keyword distinction

In this section, we carry out experiments on adding weights to distinguish keywords/sentences based on uncertain connections, while ASeM considered the role of seed words and uncertain keywords as equal when they contribute to aspect category representation. Table 7 shows that our approach is simpler and more efficient than prior works and optimally setting weights to preserve inherent data properties is challenging. In detail, weighting by

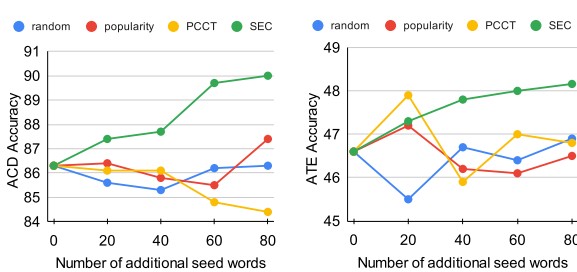

Figure 4: ACD and ATE performances with different seed word addition strategies. Our baselines: *random* adds seed words randomly from the vocabulary, mapped to their aspects by Eq.2, while *popularity* selects the most frequently occurring important words (noun/adjective) of each aspect based on a small labeled dataset (dev set). PCCT is a set of additional seed words proposed by (Li et al., 2022), extracted from the vocabulary of a pre-trained model.

| Methods | ACD | |
| --- | --- | --- |
| | Acc | macro-F1 |
| ASeM | **90.0** | **82.0** |
| weighted term level | 88.6 | 77.5 |
| weighted sentence level | 89.1 | 75.3 |
| w/o aug + weighted sentence level | 86.4 | 74.6 |

Table 7: Experimental Results for Weighted Seed Word and Sentence Assignment on Restaurant dataset. **weighted term level**: A variant of AseM where the aspect representation is computed by a weighted sum of seed word representations (following (Angelidis and Lapata, 2018)). **weighted sentence level**: A variant of AseM where the connection score is multiplied into the loss function (following (Nguyen et al., 2021)). **w/o aug + weighted sentence level** is the weighted sentence level without data augmentation.

confidence scores does not consistently yield improved results or seed words that appear more frequently do not necessarily play a more important role, as further illustrated by Figure 4.

## 6.3 Qualitative analysis

We conducted a comparison (Table 8) of the performance improvement of our model against the UCE baseline and observed that the progress stems from two main factors. First, our model accurately identifies aspect terms present in sentences, which in turn accurately determines the ACD, as shown in rows 1 and 2 of Table 8. Our finding is further supported by Figure 4, which illustrates that the model's performance correlates with the accuracy of Aspect Term Extraction (ATE). Additionally,

| Sentences | UCE | ASeM |
| --- | --- | --- |
| Save *room* for scrumptious **desserts**. | ambience | food |
| This *place* is famous for their **breakfast**. | location | food |
| The **waiters** are very **experienced** and helpful with pairing your *drink* choice to your *food* tastes or vice versa. | food | service |
| Can't believe how an **expensive** *NYC* restaurant can be so **disrespectful** to its clients. | location | service |

Table 8: Examples of improved accuracy by ASeM

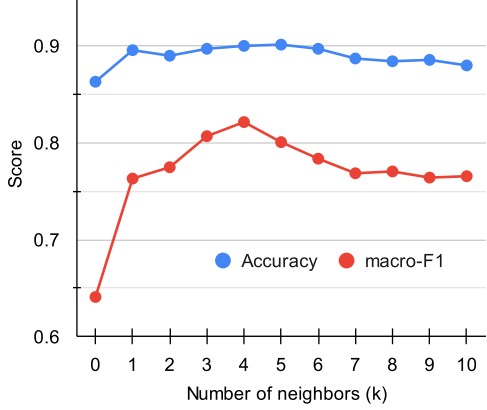

Figure 5: Performance of ACD with an increasing number of neighbors. More neighbors result in a larger training data set.

our model accurately associates the sentiment of a sentence with the corresponding aspect, thereby enhancing ACD performance as demonstrated by rows 3 and 4 of Table 8.

## 6.4 Retrieval-based Augmentation

In this subsection, we examine the impact of transforming the unsupervised learning problem based on seed words into a data augmentation task for a low-resource task. As observed in Figure 5, our framework's performance shows a substantial improvement in the initial phase but gradually declines afterward. This decline can be attributed to an excessive increase in neighbors, which leads to the inclusion of misaligned data that does not connect well with the target task's prior knowledge (e.g. seed words). Consequently, the pseudo-label generation becomes insufficient to provide accurate predictions, resulting in compromised classification and decreased performance.

## 7 Conclusion

In this work, we propose a novel framework for ACD that achieves three main goals: (1) enhancing aspect understanding and reducing reliance on initial seed words, (2) effectively handling noise in the training data, and (3) self-boosting supervised signals through multitask-learning three unsupervised tasks (ACD, ATE, ATP) to improve performance. The experimental results demonstrate that our model outperforms the baselines and achieves state-of-the-art performance on three benchmark datasets. In the future, we plan to extend our framework to address other unsupervised problems.

## Limitations

Although our experiments have proven the effectiveness of our proposed method, there are still some limitations that can be improved in future work. First, our process of assigning keywords to their relevant aspects is not entirely accurate. Future work may explore alternatives to make this process more precise. Second, through the analysis of the results, we notice that our framework predicts the aspect categories of sentences with implicit aspect terms less accurately than sentences with explicit aspect terms. This is because we prioritize the presence of aspect terms in sentences when predicting their aspect categories, which can be seen in the pseudo-label generation. However, sentences with implicit aspect terms do not contain aspect terms, or even contain terms of other aspects, leading to incorrect predictions. For example, *the only beverage we did receive was water in dirty glasses* was predicted as DRINKS instead of the golden aspect label SERVICE. Future works may focus more on the context of sentences to make better predictions for sentences with implicit aspect terms.

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

## A   GPT-3.5 prompt

In the experiment with GPT-3.5, we use the following prompt:

> Map the following sentence to the appropriate aspect from the provided list of n aspects:
> Sentence: {sentence}
> Aspect options: {List of n aspects}
> For each aspect, you have the following prior knowledge words:
> {Aspect 1}: {List of seed words}
> ...
> {Aspect n}: {List of seed words}
> Your task is to associate the sentence

with the most suitable aspect using the
provided prior knowledge.