# OpenReview forum: "A Self-enhancement Multitask Framework for Unsupervised Aspect Category Detection"
_EMNLP/2023/Conference — EMNLP 2023 Main_

### Official Review · Reviewer_UyNx · 2023-08-03

**Soundness:** 3

**Excitement:**

3: Ambivalent: It has merits (e.g., it reports state-of-the-art results, the idea is nice), but there are key weaknesses (e.g., it describes incremental work), and it can significantly benefit from another round of revision. However, I won't object to accepting it if my co-reviewers champion it.

**Paper Topic And Main Contributions:**

The paper presents ASeM, an unsupervised framework for Aspect Category Detection (ACD). ASeM addresses issues related to the quality of initial seed words and sample noise in the dataset based on previous and recent works. The contributions of the paper include enhancing aspect comprehension and reducing reliance on the quality of initial seed words, introducing retrieval-based data augmentation to handle training data noise, leveraging multi-task learning for improved ACD, and demonstrating superior performance compared to previous methods through comprehensive experiments on standard datasets.

**Reasons To Accept:**

1. The paper introduces the Seedword Enhancement Component (SEC) that expands the set of seed words to enhance their quality. Then, ASeM shows good adaptation to the quality of initial seed words during the study. Even with varying initial seed word sets, the model delivers promising results, reducing the dependency on high-quality initial seed words.

2. The experimental results demonstrate that ASeM outperforms several state-of-the-art unsupervised ACD methods on three benchmark datasets, establishing it as an effective framework in ACD tasks.

3. ​​The paper employs multitask learning for Aspect Term Extraction (ATE), Aspect Term Polarity (ATP), and Aspect Category Detection (ACD) tasks, which offer extra guidance for ACD by leveraging shared representation learning. As a result, this strategy leads to enhanced performance in all three tasks.

**Reasons To Reject:**

1. While the paper asserts that all three benchmark datasets are from different domains, it is important to note that the Restaurant and CitySearch datasets both revolve around restaurant reviews. This similarity in the domain might not be entirely convincing to demonstrate the effectiveness across different domains.

2. The paper does not mention the specific computational resources used for their experiments and the time complexity of the method. The lack of such information on the computational resources used can make it difficult for readers to assess the scalability and reproducibility of the proposed framework.

3. The performance of the ACD task heavily depends on the inconsistent quality of pseudo-label generation. The accuracy of the pseudo-labels can be impacted by the presence of noisy data or ambiguous sentences, leading to potential misclassification.

**Reproducibility:**

3: Could reproduce the results with some difficulty. The settings of parameters are underspecified or subjectively determined; the training/evaluation data are not widely available.

**Reviewer Confidence:**

4: Quite sure. I tried to check the important points carefully. It's unlikely, though conceivable, that I missed something that should affect my ratings.

---

> ### Author Rebuttal · Authors · 2023-08-28
>
> Thank you for your valuable comments. We appreciate your insights and would like to address the concerns you've raised regarding the potential reasons for rejection.
>
> > ...not be entirely convincing to demonstrate the effectiveness across different domains.
>
> **Our response**:
>
> While it's true that both the Restaurant and CitySearch datasets revolve around restaurant reviews, they exhibit variations in data sources, review styles, and aspect labels. Moreover, we have experimented with the **_Laptop_** domain, as detailed in lines 396-411, providing further evidence of our method's effectiveness across diverse domains.
>
> > The paper does not mention the specific computational resources used for their experiments and the time complexity of the method.
>
> **Our response**:
>
> The computational resources employed for our experiments included the utilization of the RoBERTa-large model, running on a Tesla V100 32GB GPU. The training duration varied based on the dataset sizes of different experiments, averaging around 10 minutes/experiment. The model employed in our work is a prevalent and extensively utilized architecture, ensuring straightforward reproducibility. Consequently, we omitted this detail from our paper.
>
> > The performance of the ACD task heavily depends on the inconsistent quality of pseudo-label generation. The accuracy of the pseudo-labels can be impacted by the presence of noisy data or ambiguous sentences, leading to potential misclassification.
>
> **Our response**:
>
> We understand that the quality of pseudo labels has an impact on the model's performance. However, our contributions are focused on tackling this matter.
> - **SEC**, outlined in lines 81-97, 142-147, and Algorithm 1, introduces enhancements to aspect comprehension and reduces dependence on the initial seed word quality. This results in improved and consistent quality of pseudo-labels with any initial seed words. Our assertion is substantiated by the compelling evidence presented in Table 5 and Figures 3, and 4.
> - **Retrieval-based Data Augmentation**, detailed in lines 97-119, 148-154, and section 3.2, is designed to automatically remove out-of-domain data and inaccurate pseudo-labels. This proactive filtering mitigates their detrimental impact on performance. Table 5 and Figure 5 provide empirical validation for our proposition.

---

### Official Review · Reviewer_9v2c · 2023-08-05

**Soundness:** 3

**Excitement:**

3: Ambivalent: It has merits (e.g., it reports state-of-the-art results, the idea is nice), but there are key weaknesses (e.g., it describes incremental work), and it can significantly benefit from another round of revision. However, I won't object to accepting it if my co-reviewers champion it.

**Paper Topic And Main Contributions:**

This paper proposes a novel framework for unsupervised aspect category detection. The task is to identify the categories, specific aspects, and polarities of the aspects discussed. The three sub-tasks can be named ACD, ATE, and ATP respectively. The authors propose two main improvements over the previous work, one is data augmentation and the other is multi-task learning. Experiments on three benchmark datasets demonstrate the effectiveness of the proposed framework.

**Reasons To Accept:**

The method is straightforward and the motivation is clear. It can achieve good performance over a few baselines included in this paper.

**Reasons To Reject:**

1. Data augmentation and multi-task learning are both well-studied techniques. I'm not surprised to apply these techniques to a new application. However, I do not see a lot of technical novelties and insights.
2. I'm not sure if LLM can help with this task. It will be interesting to see an LLM baseline.
3. It would be better to provide some qualitative analyses to discuss which types of cases could be better addressed with the new framework.

**Reproducibility:**

4: Could mostly reproduce the results, but there may be some variation because of sample variance or minor variations in their interpretation of the protocol or method.

**Reviewer Confidence:**

4: Quite sure. I tried to check the important points carefully. It's unlikely, though conceivable, that I missed something that should affect my ratings.

---

> ### Author Rebuttal · Authors · 2023-08-28
>
> We're grateful that you've recognized the efficacy of our approach. We appreciate your insights and would like to address the concerns you've raised regarding the potential reasons for rejection.
>
> >  I do not see a lot of technical novelties and insights
>
> **Our response**:
>
> **(1) Seedword enhancement component**:
> We want to emphasize that previous unsupervised ACD works have not been consistent in using the same prior knowledge (i.e. seed words). They have also shown significant variations in performance when using different sets of seed words (as observed in Figure 3). We believe that addressing this issue is crucial for better understanding how seed words affect performance and to compare ACD studies more effectively. Moreover, this Seedword Enhancement Component is executed through a meticulous analysis of the relationships between seed words and sentences, as well as among the seed words themselves (lines 83-97, Algorithm 1). As a result, it not only ensures consistent quality for generating pseudo-labels from any given seed word inputs but also enhances overall performance (as demonstrated in Table 5).
>
> **(2) Retrieval-based Data Augmentation**:
> We would like to highlight that the data augmentation technique applied to this unsupervised aspect category detection (ACD) task is unique in the following ways:
> - **Purpose of use**: Data augmentation is typically utilized for two main purposes:  (a) Increasing training data diversity without additional crawling, through techniques like synonym replacement [1,2,3]. (b) Extracting more knowledge from external sources [4,5].
> In this work, we use the data augmentation technique to **remove low-quality training data**.
> - **Application to unsupervised ACD**: We are the first to approach the challenge of addressing noise in unsupervised ACD as a data augmentation problem. Unlike previous unsupervised ACD methods that train on all available unlabeled data while trying to limit the harm of inaccurate pseudo-labels based on a certain criteria (e.g. confidence of classification model [6]), we extract a small subset from the training data to act as a task-specific in-domain data. We then proceed to collect more high-quality data from the rest through its connection with seed words by the data-augmentation technique. (line 221-235, sub-section 3.5)
>
> Intuition: In-domain sentences predictable from the seed words have strong connections in content and context to those seed words. In contrast, out-of-domain sentences and those the model cannot accurately predict from the seed words often contain irrelevant or minimal relevant information. Thus, using a data augmentation technique in this way simultaneously increases useful in-domain data, while preventing an increase of irrelevant examples and inaccurate pseudo-labels.
>
> Our experiments demonstrate that our approach is both simpler and more efficient than previous studies. This is because determining the right weights to maintain the natural characteristics of the data is a complex task. For instance, seed words that occur more often might not always be more significant (see Figure 4).
>
> > I'm not sure if LLM can help with this task. It will be interesting to see an LLM baseline.
>
> **Our response**:
>
> We highly appreciate your suggestions, and we also agree that incorporating additional concerns with state-of-the-art Large Language Models (LLMs) would enhance the significance of our research and open up new avenues for future exploration. We will include the following discussion in the revised version.
>
> First and foremost, we want to emphasize that the Roberta model we are utilizing serves as a foundational and reasonable LLM, particularly in ensuring fairness with related works and minimal computational resource requirements. It offers ease of expansion and reuse.
>
> Regarding the extension of using larger LLMs, there are several distinct approaches:
> - Concerning the utilization of seed words to guide inference for larger LLMs, such as generative models like GPT-3.5, we conducted an experiment involving GPT-3.5* (from OpenAI) to infer aspect labels. The outcomes are detailed in Table 7.
>
> | **Method** | **FOOD** | **DRINKS** | **SERVICE** | **LOCATION** | **AMBIENCE** | **Overall** |  |
> |:---:|:---:|:---:|:---:|:---:|:---:|:---:|:---:|
> |  | **F1** | **F1** | **F1** | **F1** | **F1** | **Acc** | **Macro_f1** |
> | ASeM | 93.4 | 76.2 | 87.7 | **63.2** | **88.4** | 90.0 | **82.0** |
> | GPT-3.5 | **94.9** | **84.7** | **91.2** | 45.5 | 85.0 | **91.4** | 80.3 |
>
> Table 7 Performance of ASeM vs. GPT-3.5 on Restaurant Dataset.
>
> As can be observed, while GPT-3.5 performs well in terms of food, drinks, and service aspects and demonstrates superior accuracy, our model outperforms in terms of location and ambience aspects as well as overall macro-F1 score.
>
> It's not surprising that GPT-3.5 demonstrates strong performance on this dataset due to its immense underlying knowledge base. However, when considering factors like scalability, computational demands, complexity, and inference time, our method exhibits competitive results.
>
> **\* _In this experiment, we use the following prompt:_**
>
> ```
> Map the following sentence to the appropriate aspect from the provided list of n aspects:
> Sentence: [Insert sentence here]
> Aspect Options: [List of n aspects]
> For each aspect, you have the following prior knowledge words:
> [Aspect 1]: [List of seed words for Aspect 1]
> [Aspect 2]: [List of seed words for Aspect 2]
> ...
> [Aspect n]: [List of seed words for Aspect n]
> Your task is to associate the sentence with the most suitable aspect using the provided prior knowledge.
> ```
>
> - As a potential avenue for future work, similar to our existing approach of weakly supervised learning, the prospect of substituting Roberta with a more robust LLM offers the potential for significant improvements.
>
>
> > It would be better to provide some qualitative analyses to discuss which types of cases could be better addressed with the new framework
>
> **Our response**:
>
> We agree that adding qualitative analyses would provide further clarity on the instances where our model outperforms. We will incorporate these analyses into the revised version.
>
> We conducted a comparison of the performance improvement of our model against the UCE baseline (Table 8) and observed that the progress stems from two main factors. First, our model accurately identifies aspect terms present in sentences, which in turn accurately determines the ACD, as shown in rows 2 and 3 of Table 8. Our finding is further supported by Figure 4, which illustrates that the model's performance correlates with the accuracy of Aspect Term Extraction (ATE). Additionally, our model accurately associates the sentiment of a sentence with the corresponding aspect, thereby enhancing ACD performance as demonstrated by rows 4 and 5 of Table 8.
>
> | **Sentences** | **UCE** | **ASeM** |
> |---|---|---|
> | Save _room_ for scrumptious **desserts**. | ambience | food |
> | This _place_ is famous for their **breakfast**. | location | food |
> | The **waiters** are very **experienced** and helpful with pairing your _drink_ choice to your _food_ tastes or vice versa. | food | service |
> | Can’t believe how an **expensive** _NYC_ restaurant can be so **disrespectful** to its clients. | location | service |
>
> Table 8: Examples of Improved Accuracy by ASeM
>
>
> Citations:
>
> [1] Asai, A. and Hajishirzi, H., 2020, July. Logic-Guided Data Augmentation and Regularization for Consistent Question Answering. In Proceedings of the 58th Annual Meeting of the Association for Computational Linguistics (pp. 5642-5650).
>
> [2] Wei, J. and Zou, K., 2019, November. EDA: Easy Data Augmentation Techniques for Boosting Performance on Text Classification Tasks. In Proceedings of the 2019 Conference on Empirical Methods in Natural Language Processing and the 9th International Joint Conference on Natural Language Processing (EMNLP-IJCNLP) (pp. 6382-6388).
>
> [3] Wei, J., Huang, C., Vosoughi, S., Cheng, Y. and Xu, S., 2021, June. Few-Shot Text Classification with Triplet Networks, Data Augmentation, and Curriculum Learning. In Proceedings of the 2021 Conference of the North American Chapter of the Association for Computational Linguistics: Human Language Technologies (pp. 5493-5500).
>
> [4] Du, J., Grave, É., Gunel, B., Chaudhary, V., Celebi, O., Auli, M., Stoyanov, V. and Conneau, A., 2021, June. Self-training Improves Pre-training for Natural Language Understanding. In Proceedings of the 2021 Conference of the North American Chapter of the Association for Computational Linguistics: Human Language Technologies (pp. 5408-5418).
>
> [5] Xia, M., Kong, X., Anastasopoulos, A. and Neubig, G., 2019, July. Generalized Data Augmentation for Low-Resource Translation. In Proceedings of the 57th Annual Meeting of the Association for Computational Linguistics (pp. 5786-5796).
>
> [6] Huang, J., Meng, Y., Guo, F., Ji, H. and Han, J., 2020, November. Weakly-Supervised Aspect-Based Sentiment Analysis via Joint Aspect-Sentiment Topic Embedding. In Proceedings of the 2020 Conference on Empirical Methods in Natural Language Processing (EMNLP) (pp. 6989-6999).
>
> [7] Lei Zhang and Bing Liu. 2014. Aspect and Entity Extraction for Opinion Mining, pages 1–40. Springer Berlin Heidelberg, Berlin, Heidelberg.
>
> [8] Stefanos Angelidis and Mirella Lapata. 2018. Summarizing opinions: Aspect extraction meets sentiment prediction and they are both weakly supervised. In Proceedings of the 2018 Conference on Empirical Methods in Natural Language Processing, pages 3675–3686, Brussels, Belgium. Association for Computational Linguistics.
>
> [9] Karamanolakis, G., Mukherjee, S., Zheng, G. and Hassan, A., 2021, June. Self-Training with Weak Supervision. In Proceedings of the 2021 Conference of the North American Chapter of the Association for Computational Linguistics: Human Language Technologies (pp. 845-863).
>
> [10] Stefanos Angelidis and Mirella Lapata. 2018. Summarizing opinions: Aspect extraction meets sentiment prediction and they are both weakly supervised. In Proceedings of the 2018 Conference on Empirical Methods in Natural Language Processing, pages 3675–3686, Brussels, Belgium. Association for Computational Linguistics
>
> [11] Thi-Nhung Nguyen, Kiem-Hieu Nguyen, Young-In Song, and Tuan-Dung Cao. 2021. An Uncertainty-Aware Encoder for Aspect Detection. In Findings of the Association for Computational Linguistics: EMNLP 2021, pages 797–806, Punta Cana, Dominican Republic. Association for Computational Linguistics.

---

### Official Review · Reviewer_3mKJ · 2023-08-10

**Soundness:** 4

**Excitement:**

3: Ambivalent: It has merits (e.g., it reports state-of-the-art results, the idea is nice), but there are key weaknesses (e.g., it describes incremental work), and it can significantly benefit from another round of revision. However, I won't object to accepting it if my co-reviewers champion it.

**Missing References:**

None

**Paper Topic And Main Contributions:**

This paper proposes a method to tackle the problem of seed word selection in unsupervised aspect category detection. The proposed method adds new keywords with uncertain connections to the initial seed word set (Pseudo-label generation). Secondly, it leverages a retrieval-based data augmentation technique to search for augmented training data in unannotated data banks. Lastly, it deploys multi-task learning with tasks having similar nature, e.g., aspect term extraction and aspect term polarity tasks, which are also trained based on unsupervised pseudo-label generation.

**Questions For The Authors:**

a) Does your method consider the difference between keywords with certain connections and keywords with uncertain connections? Do you think adding weights to distinguish keywords from uncertain connections (similar to Huang et al., 2020; Nguyen et al., 2021 mentioned at 072) can help the learning, or have you experimented with that?

**Reasons To Accept:**

The authors provide clear explanations about the methodology. The writing is easy to understand. The proposed method is a reasonable solution to the specific mentioned problem (unsupervised selection of keywords in aspect category detection). Aside from the overall accuracy and macro-F1 (outperforms existing method with large margins), they also provide many additional analyses and experiments which are important evidence for their claim (e.g., with more keywords selected by the proposed method, the accuracy increased).

**Reasons To Reject:**

The main contribution of this paper is a solution to a very specific sub-problem, which is expanding the keyword set for unsupervised aspect category detection, of aspect sentiment analysis. And the problem isn’t unique to the aspect category detection. There are similar methods in other unsupervised learning tasks. Therefore, the impact of this paper might be limited.

**Reproducibility:**

4: Could mostly reproduce the results, but there may be some variation because of sample variance or minor variations in their interpretation of the protocol or method.

**Reviewer Confidence:**

4: Quite sure. I tried to check the important points carefully. It's unlikely, though conceivable, that I missed something that should affect my ratings.

**Typos Grammar Style And Presentation Improvements:**

a) It would be easier to read if the first paragraph in section one can contain a direct description of Table 1.

---

> ### Author Rebuttal · Authors · 2023-08-28
>
> Thank you for acknowledging the effectiveness of our method and the thoroughness of the experiments that support our claims. We appreciate your insights and would like to address the concerns you've raised regarding the potential reasons for rejection.
>
> > There are similar methods in other unsupervised learning tasks....
>
> **Our Response**:
>
> **(1) Seedword enhancement component**:
> We want to emphasize that previous unsupervised ACD works have not been consistent in using the same prior knowledge (i.e. seed words). They have also shown significant variations in performance when using different sets of seed words (as observed in Figure 3). We believe that addressing this issue is crucial for better understanding how seed words affect performance and to compare ACD studies more effectively. Moreover, this Seedword Enhancement Component is executed through a meticulous analysis of the relationships between seed words and sentences, as well as among the seed words themselves (lines 83-97, Algorithm 1). As a result, it not only ensures consistent quality for generating pseudo-labels from any given seed word inputs but also enhances overall performance (as demonstrated in Table 5).
>
> **(2) Retrieval-based Data Augmentation**:
> We would like to highlight that the data augmentation technique applied to this unsupervised aspect category detection (ACD) task is unique in the following ways:
> - **Purpose of use**: Data augmentation is typically utilized for two main purposes:  (a) Increasing training data diversity without additional crawling, through techniques like synonym replacement [1,2,3]. (b) Extracting more knowledge from external sources [4,5].
> In this work, we use the data augmentation technique to **remove low-quality training data**.
> - **Application to unsupervised ACD**: We are the first to approach the challenge of addressing noise in unsupervised ACD as a data augmentation problem. Unlike previous unsupervised ACD methods that train on all available unlabeled data while trying to limit the harm of inaccurate pseudo-labels based on a certain criteria (e.g. confidence of classification model [6]), we extract a small subset from the training data to act as a task-specific in-domain data. We then proceed to collect more high-quality data from the rest through its connection with seed words by the data-augmentation technique. (line 221-235, sub-section 3.5)
>
> Intuition: In-domain sentences predictable from the seed words have strong connections in content and context to those seed words. In contrast, out-of-domain sentences and those the model cannot accurately predict from the seed words often contain irrelevant or minimal relevant information. Thus, using a data augmentation technique in this way simultaneously increases useful in-domain data, while preventing an increase of irrelevant examples and inaccurate pseudo-labels.
>
> >  the impact of this paper might be limited
>
> **Our Response**:
>
> As this problem extends beyond ACD, our techniques could be adapted to aspect-based sentiment analysis [7], summarization [8], and related weakly supervised tasks [9] where managing noise and finding supervised signals are common challenges.
>
> > ...difference between keywords...Do you think adding weights to distinguish keywords...
>
> **Our Response**:
>
> We considered the role of seed words and uncertain keywords as equal (line 252). Our in-house experiments about adding weights to distinguish keywords/sentences are reported in Table 6.
>
> |            **Methods**            |  **ACD** |              |
> |:--------------------------------- |--------: | ------------:|
> |                                   | **Acc**  | **macro-F1** |
> | ASeM                              | **90.0** | **82.0**     |
> | weighted term level               | 88.6     | 77.54        |
> | weighted sentence level           | 89.1     | 75.3         |
> | w/o aug + weighted sentence level | 86.4     | 74.6         |
>
> Table 6. Experimental Results for Weighted Seed Word and Sentence Assignment on Restaurant dataset.  **weighted term level**: A variant of AseM where the aspect representation is computed by a weighted sum of seed word representations (following [10]). **weighted sentence level**: A variant of AseM where the connection score is multiplied into the loss function (following [11]). **w/o aug + weighted sentence level** is the weighted sentence level without data augmentation.
>
> The results show that our approach is simpler and more efficient than previous studies because optimally setting weights to preserve inherent data properties is challenging. As Figure 4 shows, seed words that appear more frequently do not necessarily play a more important role. Additionally, Table 6 demonstrates that weighting by confidence scores does not consistently yield improved results. Setting weights to distinguish seed words or pseudo-label quality remains an open challenge. We will expand on this issue in a revised version of the paper.
>
> > It would be easier to read if the first paragraph in section one can contain a direct description of Table 1.
>
> **Our response**:
>
> Thank you for your suggestion. In the revised version, we will include a direct description of Table 1 in the first paragraph of section 1 for improved readability.
>
> Citations:
>
> [1] Asai, A. and Hajishirzi, H., 2020, July. Logic-Guided Data Augmentation and Regularization for Consistent Question Answering. In Proceedings of the 58th Annual Meeting of the Association for Computational Linguistics (pp. 5642-5650).
>
> [2] Wei, J. and Zou, K., 2019, November. EDA: Easy Data Augmentation Techniques for Boosting Performance on Text Classification Tasks. In Proceedings of the 2019 Conference on Empirical Methods in Natural Language Processing and the 9th International Joint Conference on Natural Language Processing (EMNLP-IJCNLP) (pp. 6382-6388).
>
> [3] Wei, J., Huang, C., Vosoughi, S., Cheng, Y. and Xu, S., 2021, June. Few-Shot Text Classification with Triplet Networks, Data Augmentation, and Curriculum Learning. In Proceedings of the 2021 Conference of the North American Chapter of the Association for Computational Linguistics: Human Language Technologies (pp. 5493-5500).
>
> [4] Du, J., Grave, É., Gunel, B., Chaudhary, V., Celebi, O., Auli, M., Stoyanov, V. and Conneau, A., 2021, June. Self-training Improves Pre-training for Natural Language Understanding. In Proceedings of the 2021 Conference of the North American Chapter of the Association for Computational Linguistics: Human Language Technologies (pp. 5408-5418).
>
> [5] Xia, M., Kong, X., Anastasopoulos, A. and Neubig, G., 2019, July. Generalized Data Augmentation for Low-Resource Translation. In Proceedings of the 57th Annual Meeting of the Association for Computational Linguistics (pp. 5786-5796).
>
> [6] Huang, J., Meng, Y., Guo, F., Ji, H. and Han, J., 2020, November. Weakly-Supervised Aspect-Based Sentiment Analysis via Joint Aspect-Sentiment Topic Embedding. In Proceedings of the 2020 Conference on Empirical Methods in Natural Language Processing (EMNLP) (pp. 6989-6999).
>
> [7] Lei Zhang and Bing Liu. 2014. Aspect and Entity Extraction for Opinion Mining, pages 1–40. Springer Berlin Heidelberg, Berlin, Heidelberg.
>
> [8] Stefanos Angelidis and Mirella Lapata. 2018. Summarizing opinions: Aspect extraction meets sentiment prediction and they are both weakly supervised. In Proceedings of the 2018 Conference on Empirical Methods in Natural Language Processing, pages 3675–3686, Brussels, Belgium. Association for Computational Linguistics.
>
> [9] Karamanolakis, G., Mukherjee, S., Zheng, G. and Hassan, A., 2021, June. Self-Training with Weak Supervision. In Proceedings of the 2021 Conference of the North American Chapter of the Association for Computational Linguistics: Human Language Technologies (pp. 845-863).
>
> [10] Stefanos Angelidis and Mirella Lapata. 2018. Summarizing opinions: Aspect extraction meets sentiment prediction and they are both weakly supervised. In Proceedings of the 2018 Conference on Empirical Methods in Natural Language Processing, pages 3675–3686, Brussels, Belgium. Association for Computational Linguistics
>
> [11] Thi-Nhung Nguyen, Kiem-Hieu Nguyen, Young-In Song, and Tuan-Dung Cao. 2021. An Uncertainty-Aware Encoder for Aspect Detection. In Findings of the Association for Computational Linguistics: EMNLP 2021, pages 797–806, Punta Cana, Dominican Republic. Association for Computational Linguistics.

---

### Meta-Review · Area_Chair_t28A · 2023-09-17

**Recommendation:** 5

**Metareview:**

The reviewers are in general agreement that this paper could be accepted in EMNLP-2023. The reviewers also raised some concerns about this submission. The authors have provided a rebuttal that appeared to alleviate some concerns. A suggestion was made to accept the paper as Main Conference. The reviewers' concerns are also suggested to be addressed in the final version.

---

### Decision · Program_Chairs · 2023-10-07

**Decision:**

Accept-Main

**Comment:**

The reviewers are in general agreement that this paper could be accepted in EMNLP-2023. The reviewers also raised some concerns about this submission. The authors have provided a rebuttal that appeared to alleviate some concerns. A suggestion was made to accept the paper as Main Conference. The reviewers' concerns are also suggested to be addressed in the final version.